# TNFSF14-Derived Molecules as a Novel Treatment for Obesity and Type 2 Diabetes

**DOI:** 10.3390/ijms221910647

**Published:** 2021-09-30

**Authors:** Mark Agostino, Jennifer Rooney, Lakshini Herat, Jennifer Matthews, Allyson Simonds, Susan E. Northfield, Denham Hopper, Markus P. Schlaich, Vance B. Matthews

**Affiliations:** 1Curtin Medical School, Curtin University, Bentley, WA 6102, Australia; mark.agostino@curtin.edu.au (M.A.); allyson.simonds@student.curtin.edu.au (A.S.); 2Curtin Health and Innovation Research Institute, Curtin University, Perth, WA 6845, Australia; 3Curtin Institute for Computation, Curtin University, Perth, WA 6845, Australia; 4Dobney Hypertension Centre, School of Biomedical Sciences—Royal Perth Hospital Unit, University of Western Australia, Perth, WA 6009, Australia; Jennifer.burchell@uwa.edu.au (J.R.); Lakshini.weerasekera@uwa.edu.au (L.H.); jen.matthews@uwa.edu.au (J.M.); 5Department of Biochemistry and Pharmacology, School of Biomedical Sciences, The University of Melbourne, Parkville, VIC 3010, Australia; susan.northfield@unimelb.edu.au (S.E.N.); dhopper@student.unimelb.edu.au (D.H.); 6School of Chemistry, Bio21 Molecular Science and Biotechnology Institute, The University of Melbourne, Parkville, VIC 3010, Australia; 7Department of Cardiology, Royal Perth Hospital, Perth, WA 6000, Australia; markus.schlaich@uwa.edu.au; 8Department of Nephrology, Royal Perth Hospital, Perth, WA 6000, Australia; 9Department of Medicine, Royal Perth Hospital, Perth, WA 6000, Australia

**Keywords:** Type 2 Diabetes, obesity, TNFSF14, LIGHT, therapy, metabolic syndrome

## Abstract

Obesity is one of the most prevalent metabolic diseases in the Western world and correlates directly with glucose intolerance and insulin resistance, often culminating in Type 2 Diabetes (T2D). Importantly, our team has recently shown that the TNF superfamily (TNFSF) member protein, TNFSF14, has been reported to protect against high fat diet induced obesity and pre-diabetes. We hypothesized that mimics of TNFSF14 may therefore be valuable as anti-diabetic agents. In this study, we use in silico approaches to identify key regions of TNFSF14 responsible for binding to the Herpes virus entry mediator and Lymphotoxin β receptor. In vitro evaluation of a selection of optimised peptides identified six potentially therapeutic TNFSF14 peptides. We report that these peptides increased insulin and fatty acid oxidation signalling in skeletal muscle cells. We then selected one of these promising peptides to determine the efficacy to promote metabolic benefits in vivo. Importantly, the TNFSF14 peptide 7 reduced high fat diet-induced glucose intolerance, insulin resistance and hyperinsulinemia in a mouse model of obesity. In addition, we highlight that the TNFSF14 peptide 7 resulted in a marked reduction in liver steatosis and a concomitant increase in phospho-AMPK signalling. We conclude that TNFSF14-derived molecules positively regulate glucose homeostasis and lipid metabolism and may therefore open a completely novel therapeutic pathway for treating obesity and T2D.

## 1. Introduction

Obesity in adults can result in adverse metabolic conditions including inflammation, dyslipidemia, glucose intolerance, insulin resistance and Type 2 Diabetes (T2D) [1]. The rising tide of obesity and Type 2 Diabetes (T2D) is having a profound economic and social cost to the community [2]. This group of pathologies, which are characteristic of obesity-dependent T2D or ‘diabesity’, are beginning to emerge in children [3,4,5]. Therefore, there is an urgent need for new therapeutic options for diabesity. Current treatment options include lifestyle changes such as reduced caloric intake, increased exercise and a healthy diet. If these interventions fail, then bariatric surgery or drug therapy are viable options. Disappointingly, none of the commonly used anti-obesity drugs have resulted in consistent and effective weight loss [6] and as such there is great interest in developing new therapies to reduce obesity and its common detrimental consequence of T2D.

Numerous secreted factors have been implicated in the aetiology of diabesity in both humans and rodents. One option for drug therapy is soluble pro-inflammatory cytokines including interleukin-6 (IL-6) and granulocyte macrophage colony stimulating factor (GM-CSF). These cytokines have exhibited profound beneficial metabolic effects such as the promotion of fat oxidation and insulin sensitivity and reductions in food intake and body weight, respectively [7,8]. The adipokines—adiponectin and leptin—may also have a positive effect on body weight regulation [9,10].

Another soluble factor of interest in obesity and T2D is the Tumor Necrosis Factor (TNF) superfamily member, TNFSF14. The role of TNFSF14 in the metabolic syndrome was controversial initially. Serum TNFSF14 levels are increased in morbidly obese humans [11] and expression of TNFSF14 is reduced in non-T2D patients compared with T2D patients. These observations raise the question as to whether the upregulation of TNFSF14 during obesity is acting in a pro- or anti-obesogenic manner. TNFSF14 is highly expressed in multiple immune cells, including resting and activated T cells, B cells, monocytes and macrophages [12]. Interestingly, treatment of human primary adipocytes with TNFSF14 resulted in a potent inhibition of adipocyte differentiation and accumulation, suggesting that TNFSF14 may protect against obesity [13]. However, the exact role that TNFSF14 plays in the metabolic syndrome is unknown. We have also recently published that ablation of TNFSF14 in vivo promotes high fat diet-induced obesity, glucose intolerance, insulin resistance, hyperinsulinemia, liver steatosis and adipocyte hypertrophy and inflammation [14].

A number of recent studies have demonstrated the metabolic benefits of small peptides [15,16]. A glucagon-like receptor agonist, Semaglutide, has shown cardioprotective effects in addition to weight loss in patients with T2D [17]. Similarly, a promising role of the natural peptide Glucagon-Like Peptide 1 (GLP-1) in murine studies of obesity [18] has been documented. GLP-1 is a small peptide of 30 amino acids that has been shown to be metabolically active [18]. Therefore, we sought to synthesize similar sized peptides in relation to our target of interest, TNFSF14. In this study, we have synthesized six novel TNFSF14-derived peptides with predicted improved affinity, solubility and fold stability. The peptides were then assessed for their effects on insulin and fatty acid oxidation signalling in vitro. Peptide 7 was selected for in vivo studies as it was shown to consistently increase insulin sensitivity at both 24 and 48 h post-treatment in our cellular studies. 

In summary, we have successfully synthesized six new TNFSF14 peptides and tested their effect on the metabolic syndrome in a high fat diet mouse model. From preliminary in vitro results, one TNFSF14 peptide, Peptide 7, was chosen to determine the in vivo metabolic benefits. Peptide 7 reduced high fat diet-induced glucose intolerance, insulin resistance, hyperinsulinemia and liver steatosis. Therefore, we conclude that TNFSF14-derived molecules may improve glucose homeostasis and lipid metabolism and may therefore open a completely novel therapeutic pathway for treating obesity and T2D.

## 2. Results

### 2.1. Identification of TNFSF14 Regions Contributing to Receptor Binding

We prepared homology models of the mouse TNFSF14 ligand in complex with its receptors, TNFRSF14 and lymphotoxin-β receptor (LTβR), based on the crystal structures of the corresponding human complexes (PDBs 4RSU and 4MXW). Computational alanine scanning of the ligand identified three regions of TNFSF14 contributing the most to interaction with TNFRSF14 and LTβR (Figure 1; Table 1). 

### 2.2. TNSFS14 Peptide-2 Promotes Insulin Sensitivity In Vitro

L6 skeletal muscle myotubes were either left untreated, or treated with Insulin and Vehicle, or Insulin and TNFSF14 peptide for 24–72 h. Insulin increased the expression of phospho-AKT when compared to Vehicle-treated cells. Cells that were treated with Insulin and TNFSF14 peptide-2 demonstrated an elevation in pAKT expression when compared with cells treated with Insulin and Vehicle alone (Figure 2). 

### 2.3. Optimisation of the TNFSF14 Peptide-2

Having demonstrated that TNFSF14 Peptide 2 possesses TNFSF14-mimetic activity, design efforts focussed on optimising this peptide for improved fold stability, receptor binding and physicochemical properties. 

TNFSF14 Peptide 2 is contained within a β-hairpin region of TNFSF14, with the turn region of the β-hairpin interacting with a series of residues partially conserved between TNFRSF14 and LTβR (Figure 1C,D). Thus, it was hypothesised that preserving the β-hairpin conformation was desirable in the designed peptides. To facilitate the maintenance of β-hairpin geometry, positions at which disulfide bond-forming cysteines could be introduced were identified using the Cysteine Mutation tool of Schrodinger Biologics Suite (Table 2). Of the geometrically feasible options identified, residues 3 and 12 of peptide 2 were selected as the preferred sites for cysteine mutation, as the resulting peptide was the only one maintaining the desired β-hairpin conformation in PEP-FOLD (Appendix A). From this peptide, repeated rounds of exhaustive point mutation, affinity- and stability-based selection, aggregation checking and fold stability evaluation—as described in the Methods (In silico modelling and optimisation)—were carried out, from which six peptides were derived with differing predicted binding properties for TNFRSF14 and LTβR (Table 3).

### 2.4. Synthesis of TNFSF14 pep2 Analogues

The peptides designed from the in silico optimisation were all chemically synthesised using standard Fmoc-solid phase peptide synthesis protocols. Assembly of their linear chains proceeded smoothly with the desired peptides as the predominant product in each synthesis. Peptide oxidation to form the disulfide bonds was not trivial. A series of different oxidation conditions was trialled for this peptide series. 1.0 meq of dipyridyl disulfide was determined as the optimal condition and each of the six peptides was successfully oxidised then purified by reverse-phase high performance liquid chromatography (RP-HPLC). Peptides were purified to >95% purity for use in in vitro experiments, then characterised by liquid chromatography mass spectrometry (LCMS) (Table 4).

### 2.5. TNFSF14 Peptides Promote Insulin Signalling in L6 Skeletal Muscle Myotubes

The TNFSF14 Peptides 6, 7, 8, 9 and 10 all promoted increased insulin-induced phosphorylation of AKT (pAKT) in L6 skeletal muscle myotubes after insulin stimulation (Figure 3). Five of the six TNFSF14 peptides promoted increased insulin stimulated pAKT compared to insulin stimulation alone (Figure 3). The Peptides 6, 7, 8, 9 and 10 all significantly increased insulin sensitivity.

### 2.6. Peptides 7 and 9 Increased Fatty Acid Oxidation Signalling in Skeletal Muscle

As Peptides 7 and 9 increased insulin signalling at both 24 and 48 h post-treatment, they were used for fatty acid oxidation signalling experimentation. L6 skeletal muscle myotube cell lysates were assessed for phosphorylation of AMPKα (pAMPK) and phosphorylation of ACC (pACC) following treatment of cells with vehicle, TNFSF14 Peptide 7 or TNFSF14 Peptide 9 (Figure 4). Peptide 7 increased expression of pAMPKα (Figure 4A,B) and pACC (Figure 4A,C) and Peptide 9 produced a significant increase for pACC (Figure 4D,F). Peptides 5, 6, 8 and 10 had no effect on fatty acid oxidation signalling.

### 2.7. Peptide 7 Increased Fatty Acid Oxidation Signalling in the Liver

We chose one of the most bioactive peptides, Peptide 7, to conduct in vivo studies. There was no effect of Peptide 7 on both food intake and body weight as shown in Appendix A. We then aimed to determine if Peptide 7 had organ-specific effects. Liver lysates were assessed for expression of pAMPK following treatment with vehicle or Peptide 7 (Figure 5). In agreement with our in vitro studies, Peptide 7 increased expression of pAMPK.

### 2.8. The TNFSF14 Peptide 7 Decreased Hyperglycaemia and Glucose Intolerance In Vivo

After demonstrating that our TNFSF14 peptides promoted insulin and fatty acid oxidation signalling, we then focussed our attention on our best candidate peptide, Peptide 7, for in vivo studies where we examine glucose homeostasis. We sought to assess the effect of Peptide 7 on glucose tolerance in vivo. Mice receiving a HFD demonstrated significantly increased glucose intolerance when compared with chow-fed mice (Figure 6). The Peptide 7 administration improved glucose tolerance in the HFD fed mice (Figure 6).

### 2.9. The TNFSF14 Peptide 7 Reduced SGLT2 Expression

Sodium glucose co-transporter 2 (SGLT-2) is a major protein responsible for glucose reabsorption in the kidney and may influence glucose homeostasis. To determine the effect of Peptide 7 on expression of SGLT2, mice fed a HFD were administered vehicle or Peptide 7 for the final two weeks of high fat feeding. SGLT2 expression was examined in the kidney (Figure 7). Mice receiving Peptide 7 showed reduced luminal SGLT2 staining in proximal tubules in the kidney (Figure 7B) when compared with mice receiving vehicle (Figure 7A). The intensity of SGLT2 staining was reduced in mice receiving Peptide 7 when compared with mice receiving vehicle (Figure 7C).

### 2.10. TNFSF14 Peptide 7 Was Unable to Decrease HFD-Induced Increased Sympathetic Innervation

Tyrosine hydroxylase (TH) is routinely used as a reliable marker of sympathetic innervation which is known to influence glucose homeostasis. Here, we show that the HFD naturally activates the sympathetic nervous system (SNS) as evidenced by elevated tyrosine hydroxylase specifically in the densely innervated kidney (Figure 8) compared with normal chow fed mice. We then sought to determine the effect of Peptide 7 on HFD-induced sympathetic innervation. We discovered that Peptide 7 does not increase HFD-induced TH above HFD levels, nor does it decrease TH compared to mice fed a HFD (Figure 8).

### 2.11. Peptide 7 Does Not Alter Pro-Inflammatory Cytokine Production

Soluble pro-inflammatory cytokines have previously shown promising beneficial metabolic effects including promotion of fat oxidation and insulin sensitivity and reductions in food intake and body weight. In this study, we measured the pro-inflammatory cytokines TNF-α and IL-6 in the liver after Peptide 7 administration (Figure 9). We report that Peptide 7 administration does not reduce HFD-induced elevations in either hepatic TNF-α (Figure 9A) or IL-6 (Figure 9B). Additionally, presence of the pro-inflammatory cytokines TNF-α, IL-6 and the anti-inflammatory cytokine IL-10 did not occur in serum from mice administered Peptide 7 (data not shown).

### 2.12. Peptide 7 Reduces HFD-Induced Insulin Resistance

Insulin resistance is a hallmark of the metabolic syndrome. Therefore, we sought to analyse insulin resistance in our mouse model of obesity. Glucose levels were measured in blood from mice on a chow diet, HFD + Vehicle or HFD + Peptide 7 administration after a bolus of insulin was administered (Figure 10). Vehicle-treated mice on a HFD showed an increase in insulin resistance compared to normal chow fed mice and insulin sensitivity was improved in peptide-treated mice on a HFD.

Hyperinsulinemia is also a hallmark of insulin resistance. Insulin was measured in serum from mice on a chow diet, HFD + Vehicle or HFD + Peptide 7. Mice fed a HFD developed hyperinsulinemia, which was significantly reduced after Peptide 7 administration (Figure 11).

### 2.13. Peptide 7 Reduces HFD-Induced Liver Steatosis

As the inner mechanisms underlying NAFLD are far from being clarified [19], we assessed the effect of peptide 7 on steatosis or lipid accumulation in liver sections from mice fed a normal chow diet, HFD with vehicle or HFD with Peptide 7 administration (Figure 12). Mice fed a HFD with vehicle demonstrated increased steatosis in liver sections compared with chow-fed mice. Mice receiving HFD and Peptide 7 showed significantly reduced steatosis compared with HFD vehicle-treated mice.

## 3. Discussion

Our group has previously reported that TNFSF14 ablation exacerbates parameters of the metabolic syndrome under high fat feeding conditions, suggesting that TNFSF14 promotes metabolic benefits. It was encouraging to see that other investigators also noted that TNFSF14 reduced HFD-induced obesity and adipocyte hypertrophy [20] which solidifies our findings. However, it should be noted that one study highlighted that TNFSF14 deficiency in mice restored glucose homeostasis and reduced hepatic inflammation and steatosis [21]. In our current study, we sought to identify TNFSF14 derived peptides with therapeutic potential for the treatment of the metabolic syndrome. We have successfully synthesized six novel TNFSF14 peptides and tested them in both in vitro and in vivo settings in an effort to assess their effects on the metabolic syndrome.

A number of recent studies have elegantly described a beneficial role for TNFSF14 in limiting the harmful pathology associated with multiple inflammatory and autoimmune conditions. TNFSF14 has been reported to aid in regulating platelet aggregation and wound healing [22], in increasing the proliferation and survival of stem cells [23] and in skeletal muscle regeneration [24]. Soluble TNFSF14 may serve a protective role in multiple sclerosis [25], play a critical role in limiting disease progression and inflammation during experimental autoimmune encephalomyelitis [26] and intestinal inflammation [27]. Finally, a recent study reported that TNFSF14 promotes tumour-specific anti-tumour immunity and thus has potential as an immunotherapeutic agent to treat colon cancer [28] and liver cancer [29]. 

In our study, although the introduction of a disulfide at residues 3 and 12 of peptide 2 was predicted to result in a substantial loss in receptor binding affinity, numerous peptides based on this scaffold were identified that elicited increased insulin signalling. This anomaly can be explained by considering the likely thermodynamics of binding, as well as limitations in the ability of computational approaches to predict certain aspects of this [30]. Binding energy consists of an enthalpic (sometimes referred to as a favourable) component and an entropic (sometimes referred to as an unfavourable) component. While many computational approaches are capable of accurately predicting changes in the enthalpic component, entropic contributions are more challenging to predict [31,32] and are often outright ignored [33,34]. The predicted loss in receptor binding affinity upon mutation of residues 3 and 12 is likely to be largely associated with a reduction in the enthalpic contribution to the binding energy. However, introduction of a disulfide bond at these residues achieves a constrained peptide with a solution conformation that likely closely resembles the receptor-bound conformation. In turn, this implies that the change in entropy (or unfavourable contribution) upon receptor binding of this peptide is likely to be minimal, thus resulting in a more active peptide. Through in silico optimisation of this constrained peptide and in vitro validation, we have been able to identify new peptides that likely afford vastly improved enthalpic contributions to binding that far outweigh the enthalpy loss associated with the initial introduction of the disulfide bond into peptide 2.

A major aim of this study was to develop a peptide with improved signalling capacity and potential therapeutic efficacy. A total of six peptides were computationally identified with predicted improved affinity, solubility and fold stability. These peptides were labelled Peptide 6, Peptide 7, Peptide 8, Peptide 9 and Peptide 10 and were assessed for their effects on insulin sensitivity. Peptides were assessed at 24 and 48 h post treatment. After 24 h of treatment, only Peptide 8 significantly increased insulin signalling in skeletal muscle cells. However, 48 h after treatment, Peptides 6, 7, 9 and 10 all increased insulin signalling in skeletal muscle cells as shown by significantly increased insulin-stimulated phosphorylation of AKT when compared with insulin treatment alone (Figure 3). In addition, only the 48 h time point showed a beneficial effect of TNFSF14 peptides on fatty acid oxidation signalling as shown by an increase in pAMPK in vitro (Figure 4). These results highlight that the kinetics of peptide metabolism and the subsequent signalling may vary between the peptides suggesting that timing of therapy may be an important factor in pharmacotherapy considerations. Peptide 7 was chosen for preliminary in vivo studies as it demonstrated consistently increased insulin signalling at 24 and 48 h post-treatment and increased fatty acid oxidation signalling after 48 h in vitro.

To demonstrate the in vivo efficacy of TNFSF14 peptides in vivo, we selected one candidate, Peptide 7, which demonstrated metabolic benefits with regard to in vitro fatty acid oxidation signalling and insulin sensitivity. We then conducted in vivo studies using this candidate peptide. We first sought to assess the effect of Peptide 7 on glucose tolerance in vivo. We demonstrated increased glucose tolerance after peptide administration. We then assessed markers of sympathetic nerve innervation due to the important regulatory role that the sympathetic nervous system (SNS) plays in glucose homeostasis. High fat diet-induced sympathetic nerve innervation was not markedly decreased with the peptide.

Soluble pro-inflammatory cytokines, including interleukin-6 (IL-6) and granulocyte macrophage colony stimulating factor, have exhibited profound beneficial metabolic effects. These include promotion of fat oxidation and insulin sensitivity and reductions in food intake and body weight, respectively [7,8]. The adipokines—adiponectin and leptin—may also have a positive effect on body weight regulation [9,10]. Additionally, over-expression of interleukin-10 protected mice from diet-induced inflammation and insulin resistance in skeletal muscle [35]. Importantly, inflammation, as measured by hepatic TNF-α and IL-6, did not appear to be heightened due to the Peptide 7 administration in vivo. Additionally, presence of the pro-inflammatory cytokines TNF-α, IL-6 and the anti-inflammatory cytokine IL-10 did not occur in serum from mice administered Peptide 7. This highlights that Peptide 7 does not ignite an inflammatory cascade in vivo. If inflammation occurs following pharmacotherapy, then it is not considered a viable therapeutic option [36]. A potential future study could assess C reactive protein (CRP) levels as a marker of an acute inflammatory response. 

Future compare and contrast studies should be conducted which also include full-length TNFSF14 protein in similar in vitro and in vivo experiments as used in our current study. This will allow the metabolic benefits of the TNFSF14 peptides to be directly compared to the effects of the control full-length TNFSF14 protein.

In conclusion, we have identified six novel TNFSF14 derived peptides and tested them in an in vitro and in vivo setting as therapeutics for the metabolic syndrome. Administration of one of these in vivo, Peptide 7, reduces high fat diet-induced glucose intolerance, insulin resistance, hyperinsulinemia and liver steatosis, illustrating a promising mechanism for the treatment of diabesity and the metabolic syndrome.

## 4. Materials and Methods

### 4.1. In Silico Modelling and Optimisation

The Schrodinger Biologics Suite (version 2018-3; Schrodinger LLC, New York, NY, USA) was used for all molecular modelling, with the exception of peptide folding experiments, which were performed using the PEP-FOLD server [37].

A homology model of the mouse TNFSF14-TNFRSF14 complex was prepared against the corresponding human ligand-receptor complex (PDB 4RSU) via the heteromultimer building workflow in Prime [38,39,40]. To prepare the mouse TNFSF14-LTβR complex, the structure of the human LTβR as found in its complex with lymphotoxin-α1β2 (PDB 4MXW) [41] was first aligned to TNFRSF14 in PDB 4RSU. A sequence alignment between human LTβR and mouse LTβR was generated, then heteromultimer building used to assemble the complex. Sequence alignments used to generate mouse TNFSF14, TNFRSF14 and LTβR structures are shown in the Appendix A. For all models, the structure built consisted of the TNFSF14 trimer and a minimal number of receptor molecules with which the majority of interactions occurred. Non-template residues and residues within 6.0 Å of the ligand-receptor interfaces in each complex were subject to refinement using Prime; specifically, these residues were initially minimised, subject to the side-chain resampling, followed by an additional minimisation. Following this, all atoms in each complex were subject to energy minimisation to give the final complexes. Computational alanine scanning on TNFSF14 in each complex was performed using the Residue Scanning utility, with residues affording ΔΔG differences greater in magnitude than 5 kcal/mol selected as significant contributors to the interaction with the receptors. Alanine mutations were introduced symmetrically across the TNFSF14 trimer.

To optimise peptide 2, the Cysteine Mutation utility was used to identify potential sites where cysteine crosslinks could be reasonably introduced based on the peptide geometry [42]. PEP-FOLD was used to check that the introduced crosslinks could maintain the desired β-hairpin configuration. The Residue Scanning/Affinity Maturation utility was used to introduce exhaustive single point mutations into the disulfide-constrained peptide 2.4 (excluding the sites of the introduced crosslink) and evaluate their effect on receptor affinity and peptide stability. Residues were mutated to all proteinogenic amino acids, excluding glycine, proline, and cysteine, the introduction of which may impart undesired conformational properties and chemical reactivity (although were retained where present in the original sequence). Peptides achieving similar or improved predictions for receptor binding affinity and peptide stability (Δ Affinity and Δ Stability less than +2.0 kcal/mol) were checked for fold stability using PEP-FOLD. Peptides achieving multiple high ranked PEP-FOLD solutions in the desired β-hairpin conformation were subject to aggregation assessment using the Aggregation Surface tool [43]. PEP-FOLD-generated conformations of peptides affording AGGSCOREs at least 10% less than the peptide from which they were derived were submitted for further rounds of point mutation, affinity- and stability-based selection, fold stability evaluation and aggregation checking. Continued optimisation in this manner was performed until no further predicted improvements could be made in binding affinity without compromising solubility (i.e., reducing AGGSCORE) and fold stability.

For peptides obtained in the final rounds of the optimisation process, the Residue Scanning/Affinity Maturation utility was used to estimate the change in TNFRSF14 and LTβR binding affinities in the context of a single calculation performed for a multi-point mutation starting from the disulfide-constrained peptide 2.4. Based on this information, a small selection of peptides were chosen for synthesis covering those predicted to be selective or non-selective for either receptor.

### 4.2. Peptide Synthesis

The linear peptides required for oxidation were prepared using standard Fluorenylmethyloxycarbonyl chloride (Fmoc)-chemistry solid phase peptide synthesis protocols on 0.1 mmol scale. Peptides were synthesised on Rink Amide Resin (0.3 mmol/g loading) and assembled on a Prelude X automated peptide synthesiser. Peptide couplings used 0.45M HCTU and 20% *v/v* DIEA in DMF. Fmoc-deprotections used 20% *v/v* piperidine in DMF. The peptide-resins were washed with DMF three times between each coupling and deprotection. Peptides were cleaved from the resins using a cleavage cocktail of TFA/TIPS/H_2_O/DoDt (90:4:3:3); concentrated under a stream of nitrogen gas, then precipitated in diethyl ether. 

Crude peptides were directly oxidised using 1.0 meq of dipyridyl disulfide in a 1:1 mix of acetonitrile/water. Reaction progression was monitored using liquid chromatography mass spectrometry (LC-MS). Once completed the peptides were purified to 95% purity using reverse-phase high performance liquid chromatography (RP-HPLC) using an Agilent 1200 series HPLC system, fitted with an Eclipse XD8-C8 4.6 Å, 5 μm column. Buffer A was 0.1% *v/v* TFA in milli Q water, and buffer B was 0.08% *v/v* TFA in acetonitrile. The column eluted with a gradient of 0–60% ACN in 0.08% aqueous TFA over 10 min at a flow rate of 1 mL/min) and characterised using LC-MS (performed on an Agilent 1260 Infinity II system. The photodiode array detector HS at 215 nm coupled directly to an electrospray ionization source and an Agilent 6120 single quadrupole mass analyzer).

### 4.3. Cell Culture Experiments

Mycoplasma free L6 myoblast cells were purchased from the American Type Culture Collection (Manassas, VA, USA). Differentiation of the myoblasts was induced by transferring cells to medium containing 2% fetal calf serum when the myoblasts were ~90% confluent. Experimental treatments commenced after 7 days of differentiation when nearly all myoblasts had fused to form myotubes. On the experimental day, cells were given fresh differentiation media and treated with vehicle or TNFSF14 peptide (Peptides 5–10) for 24–72 h. Cells were then treated with or without insulin (250 ng/mL) for 30 min before cells were lysed.

### 4.4. Insulin Signalling

Insulin signalling was measured by assessing expression of phospho-AKT and phospho-AKT/Total AKT ratio in L6 skeletal muscle myotubes. Cells were administered vehicle or the TNFSF14 peptides (20 µg/mL) for 24 or 48 h prior to acute insulin stimulation [250 ng/mL]. 

### 4.5. Mice (Mus Musculus)

WT male C57BL/6 mice were obtained from the Animal Resources Centre (Perth, Australia). The mice were housed under specific pathogen-free conditions. All experiments were conducted using protocols approved by The Harry Perkins Institute for Medical Research Animal Ethics Committee (AEC172) and were conducted in accordance with the National Health and Medical Research Council of Australia Guidelines on Animal Experimentation conforming to the *Guide for the Care and Use of Laboratory Animals* [44]. 

### 4.6. Diet-Induced Obesity Mouse Model

Eight week old male C57BL6/J mice were administered either a normal chow (chow; 14.3 MJ/kg, 76% of energy from carbohydrates, 5% from fat, 19% from protein) or high fat diet (HFD; 19 MJ/kg, 35% of energy from carbohydrate, 42% from fat, 23% from protein) (Speciality Feeds, Glen Forrest, WA, Australia) for 10 weeks (*n* = 4–5 mice/group) followed by either intravenous administration of (i) vehicle or (ii) 2.5 mg/kg of TNFSF14-derived Peptide 7 molecule. Mice were reinjected every 2 days for the final 2 weeks of the dietary regiment. 

Body weights were measured weekly for all mice. Intraperitoneal glucose tolerance tests (GTT; 1 g/kg) and insulin tolerance tests (ITT; 0.5 U/kg) were performed in mice fasted for 6 h on week 11 and 12 of the diet regime respectively. After 12 weeks on their respective diets, mice were anaesthetised with isoflurane, underwent cardiac puncture to obtain blood and were euthanised by cervical dislocation. Tissues were collected and either fixed in paraformaldehyde and subsequently embedded in paraffin or snap-frozen and stored at −80 °C. 

### 4.7. Western Blotting

Rat L6 myotubes were lysed or murine liver tissue was homogenised using cytosolic extraction buffer containing phosphatase and protease inhibitors and protein concentration was quantified using protein assay solution (Bio-Rad, Hercules, CA, USA). Protein lysates were resolved by SDS-PAGE on 10% polyacrylamide gels, transferred by semi-dry transfer to PVDF membrane. Membranes were incubated overnight in primary antibody [phospho-AKT Ser473 (9271; Cell Signalling Technology Inc., Danvers, MA, USA); phospho-ACC Ser79 (3661; Cell Signalling Technology Inc., Danvers, MA, USA); phospho-AMPKα Thr172 (Rabbit mAb 2535; Cell Signalling Technology Inc., Danvers, MA, USA); mouse anti-β actin antibody (ab6276; Abcam, Cambridge, UK)] or rabbit total AMPK-α antibody (5831; Cell Signalling) using recommended dilutions. The appropriate secondary antibody was added to the membranes (either α-rabbit 800 or α-mouse 680 [LICOR] for Odyssey detection or α-mouse HRP or α-rabbit HRP [GE Healthcare Australia, Parramatta, NSW, Australia]). Detection of the relevant protein was performed via enhanced chemiluminescence (GE Healthcare) or visualised using an Odyssey detection apparatus (LICOR, Lincoln, NE, USA).

### 4.8. Haematoxylin and Eosin Staining

Freshly dissected liver was fixed using 10% Formalin and paraffin embedded. Slides containing five µm sections were dewaxed and then stained with Haematoxylin and Eosin (Sigma-Aldrich, Castle Hill, NSW, Australia). 

### 4.9. SGLT2 and TH Immunohistochemistry

Photomicrographs were taken of the kidney of each mouse using a Nikon Eclipse Ti Microscope (Nikon Instruments Inc., New York, NY, USA). For SGLT2 quantification, the intensity of proximal tubule staining in each field of view was rated against a scale of 0–3 (0 = no staining; 1 = low staining; 2 = intermediate staining; 3 = high intensity of staining). 

Sympathetic nerves were detected as punctate TH-positive nerves or nerve like fibers. To calculate TH staining in the kidney, TH-positive nerves were counted in 3 random fields of view per sample. The mean number of TH-positive nerves in each field of view for each mouse was calculated. 

### 4.10. Cytokine ELISAs

Liver tissue was homogenised in cytosolic extraction buffer containing phosphatase and protease inhibitors. Protein levels were determined using a Bradford protein assay and TNF-α and IL-6 was measured in lysates using a mouse TNF-α and IL-6 ELISA, respectively, (ELISAkit.com, Melbourne, VIC, Australia). Serum was also assessed for TNF-α, IL-6 and IL-10 ELISAs using ELISA kits (ELISAkit.com).

### 4.11. Statistical Analysis

All quantitative data are presented as mean + and/or − SD. A significance level of *p* < 0.05 was considered statistically significant. Significance was determined using Student *t*-tests (SPSS, Version 21). Graphs were generated using GraphPad Prism 7 (GraphPad Software Inc., La Jolla, CA, USA).

## Figures and Tables

**Figure 1 ijms-22-10647-f001:**
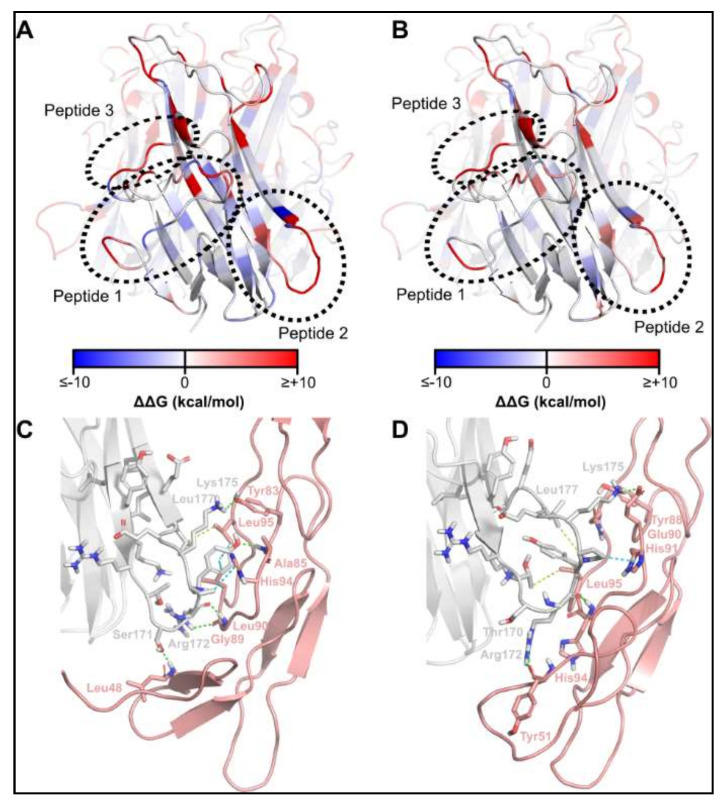
Computational alanine scanning reveals key regions of receptor contact by TNFSF14. Computational alanine scanning of TNFSF14 in complex with TNFRSF14 (**A**) and LTβR (**B**). Interaction between TNFSF14 region corresponding to peptide 2 with TNFRSF14 (**C**) and LTβR (**D**). Legend to panels **C** and **D**: grey—TNFSF14; pink—TNFRSF14/LTβR; green dashes—hydrogen bonds; cyan dashes—CH-π interactions; yellow dashes—nonpolar interactions.

**Figure 2 ijms-22-10647-f002:**
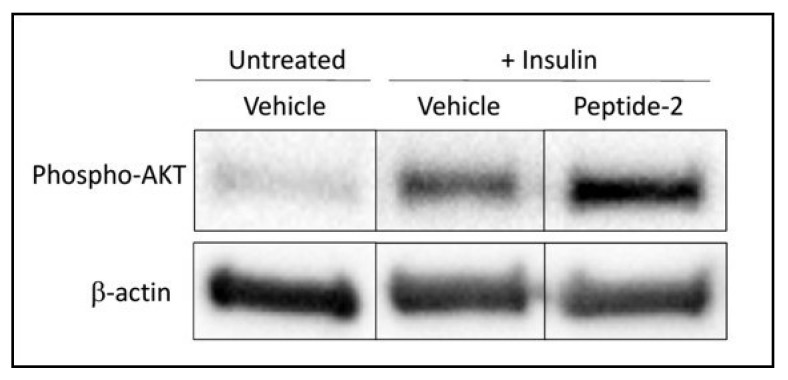
TNFSF14 Peptide 2 treatment promotes insulin sensitivity. Representative immunoblot showing Peptide 2 treatment (20 μg/mL; 48 h) promotes insulin signalling in L6 skeletal muscle myotubes as indicated by increased expression of phospho-AKT. Beta-actin served as a house keeping protein.

**Figure 3 ijms-22-10647-f003:**
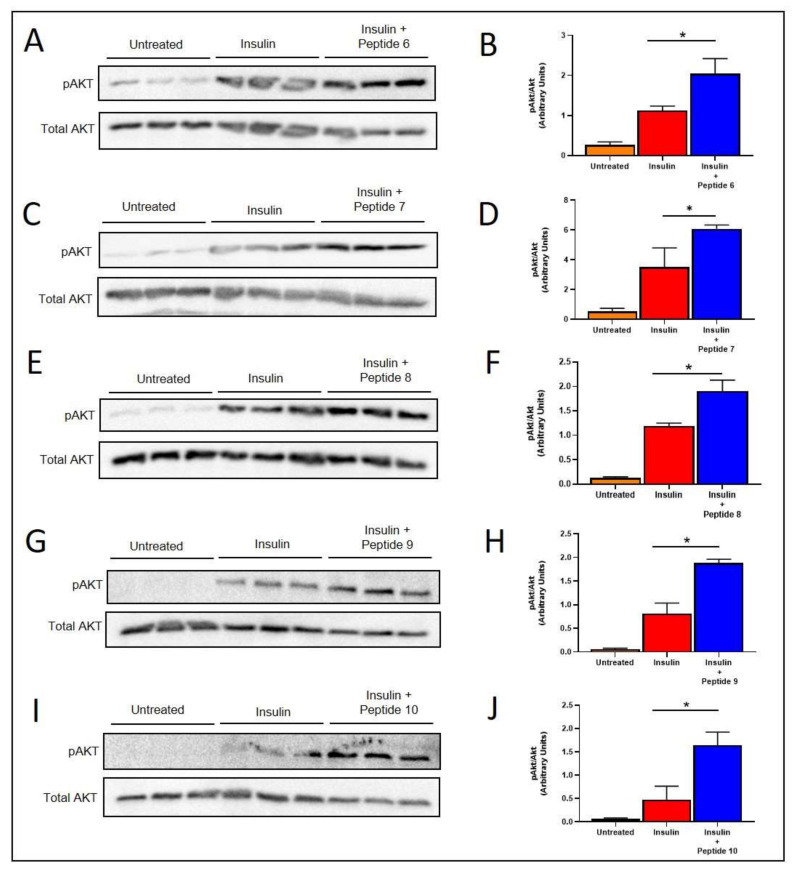
TNFSF14 peptides increase insulin signalling in skeletal muscle cells. Cells were treated with either the peptides 6 (**A**,**B**), 7 (**C**,**D**), 9 (**G**,**H**) or 10 (**I**,**J**) for 48 h prior to acute insulin stimulation. Cells were treated with Peptide 8 (**E**,**F**) for 24 h prior to acute insulin stimulation. Immunoblots for phospho-AKT and Total AKT (**A**,**C**,**E**,**G**,**I**) and quantification of pAKT/Total AKT (**B**,**D**,**F**,**H**,**J**) is shown for each peptide (*n* = 3/treatment). Mean ± SD. * *p* < 0.05. *p* = 0.02, 0.03, 0.01, 0.01 and 0.001 for (**B**,**D**,**F**,**H**,**J**) respectively.

**Figure 4 ijms-22-10647-f004:**
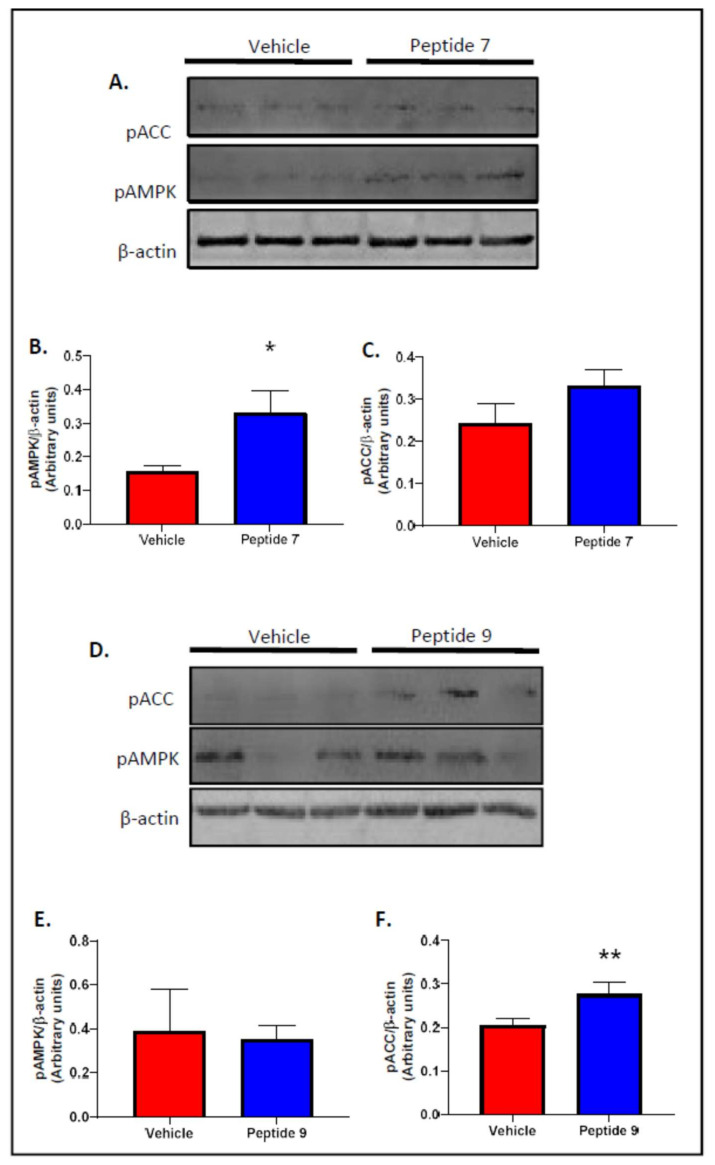
Peptides 7 and 9 increase fatty acid oxidation signalling. Cells were treated with peptide 7 or peptide 9 for 48 h. Western blots for phospho-AMPK, phospho-ACC and β-actin are shown (**A**,**D**). Quantification of phospho-AMPK (**B**,**E**) and phospho-ACC (**C**,**F**). *n* = 3/group. Mean ± SD. * *p* = 0.0176, ** *p* = 0.0167.

**Figure 5 ijms-22-10647-f005:**
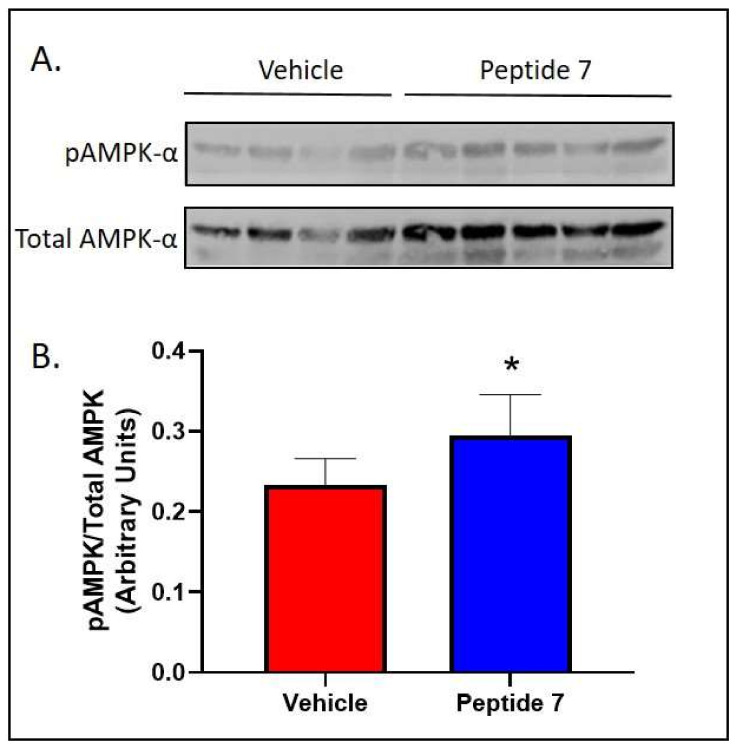
Peptide 7 administration increased fatty acid oxidation signalling in the liver. Mice were fed a HFD and treated with Peptide 7 for 2 weeks. Western blots for phospho-AMPK and Total AMPK are shown (**A**). Quantification of phospho-AMPK/Total AMPK ratio (**B**). *n* = 4–5/group. Mean ± SD. * *p* = 0.076.

**Figure 6 ijms-22-10647-f006:**
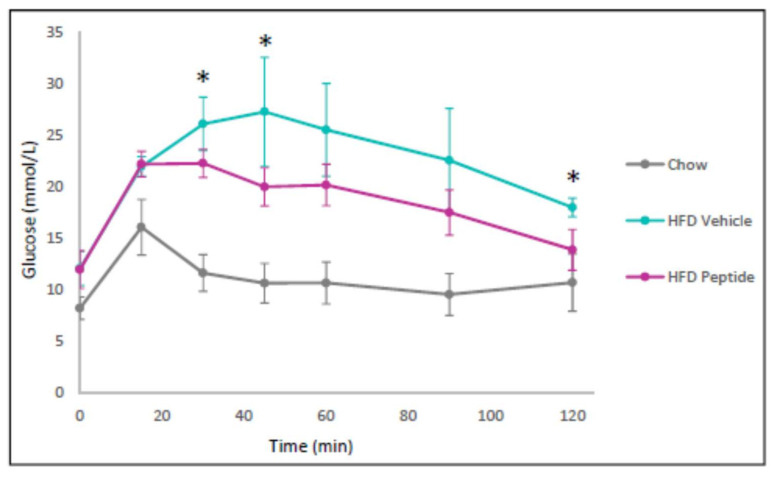
The TNFSF14 Peptide 7 reduces high fat diet-induced glucose intolerance. Mice were placed on a chow or a high fat diet (HFD) (*n* = 4–5 mice/group). The HFD was administered for 10 weeks before Vehicle or Peptide 7 therapy (2.5 mg/kg). * *p* < 0.05 (HFD vehicle versus HFD peptide). Mean ± SD.

**Figure 7 ijms-22-10647-f007:**
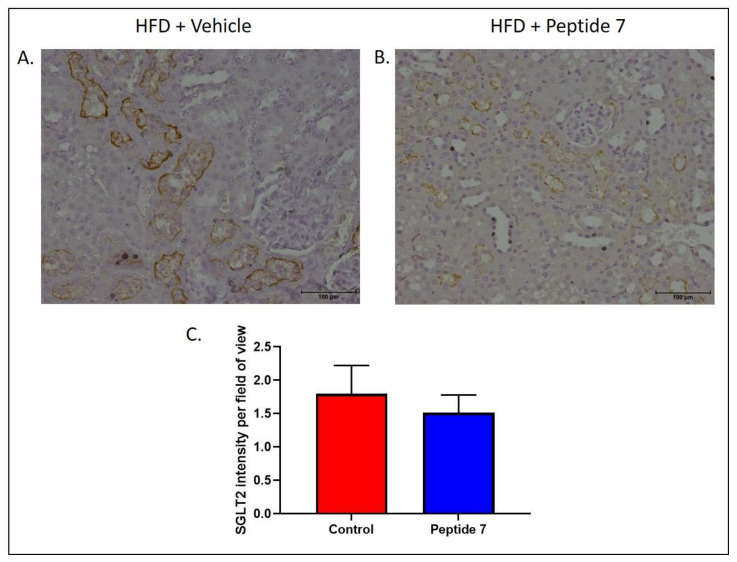
Peptide 7 reduced in vivo SGLT2 expression in HFD mice. SGLT2 expression in the kidney after treatment with vehicle or Peptide 7. Representative SGLT2 immunohistochemistry in mice fed HFD and treated with vehicle (**A**) or Peptide 7 (**B**). SGLT2 is depicted by brown staining. 200× magnification. Intensity of staining of proximal tubules in kidneys rated on a scale of 0–3, 0 = no staining, 3 = highest staining; *n* = 4–5 mice/group. Average of five random fields of view per mouse. Mean ± SD (**C**).

**Figure 8 ijms-22-10647-f008:**
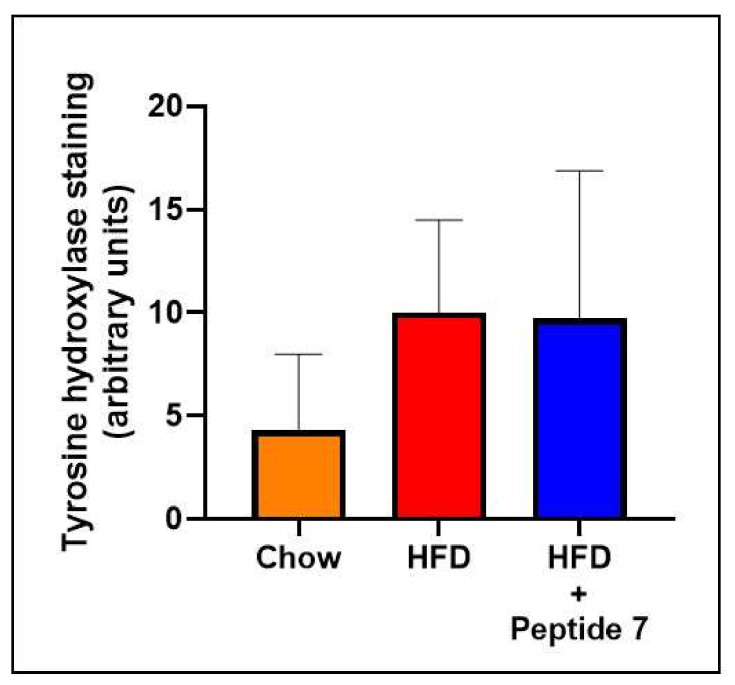
Peptide 7 did not decrease HFD-induced increases in tyrosine hydroxylase staining. Tyrosine hydroxylase expression in kidney. Tyrosine hydroxylase quantitation, *n* = 3–5 mice per group; Mean ± SD.

**Figure 9 ijms-22-10647-f009:**
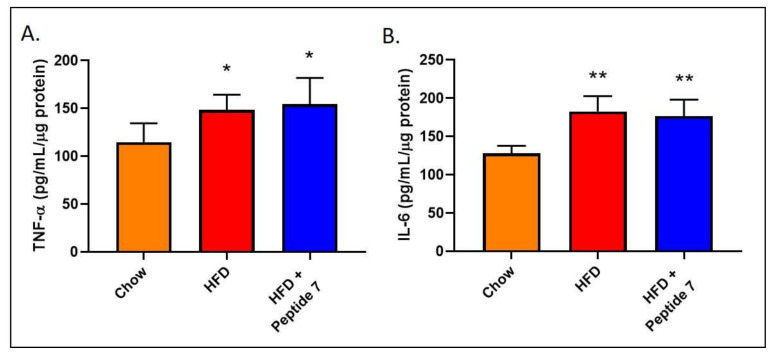
Peptide 7 does not reduce High Fat Diet-induced TNF-α (**A**) and IL-6 (**B**) in the liver. * *p* = 0.03, ** *p* = 0.002 when compared with mice on a chow diet, *n* = 4–5 mice/group with mean ± SD.

**Figure 10 ijms-22-10647-f010:**
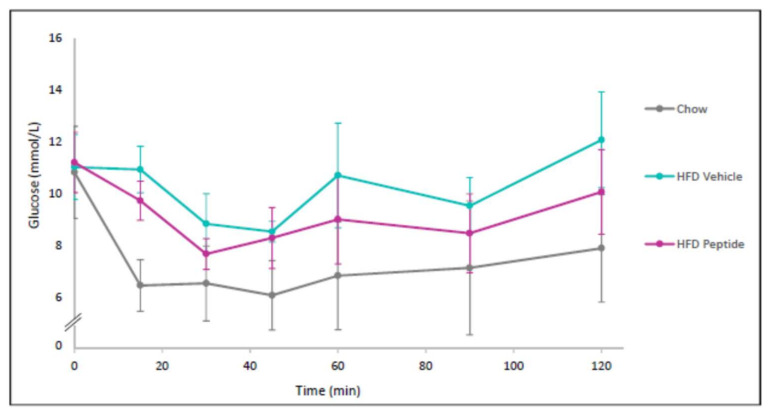
The TNFSF14 Peptide 7 reduces high fat diet-induced insulin resistance. Mice were placed on a chow or a high fat diet (HFD) (*n* = 4–5 mice/group). The HFD was administered for 10 weeks before Vehicle or Peptide 7 therapy for 2 weeks. Mean ± SD.

**Figure 11 ijms-22-10647-f011:**
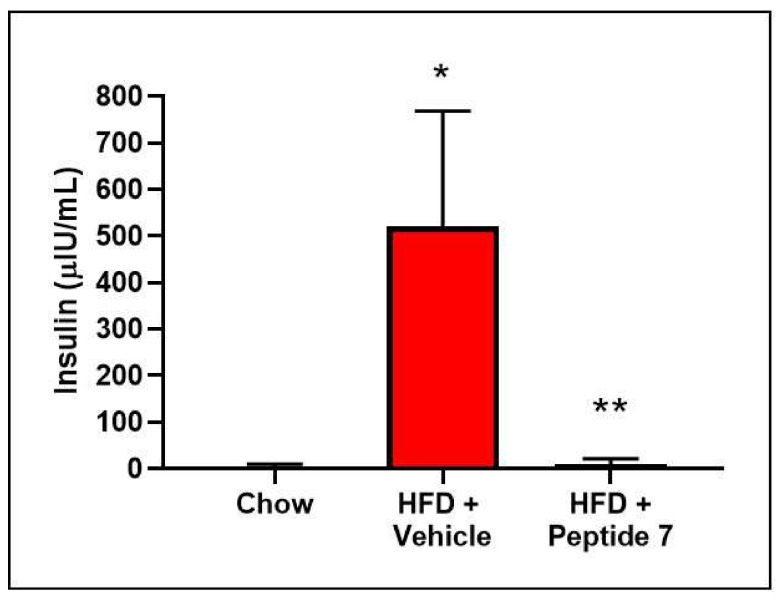
Peptide 7 significantly reduces high fat diet-induced hyperinsulinemia. Mice on a HFD for 10 weeks received either vehicle or Peptide 7 (*n* = 4–5 mice/group). * *p* = 0.0021 (compared to chow), ** *p* = 0.0023 (compared to HFD + vehicle). Mean ± SD.

**Figure 12 ijms-22-10647-f012:**
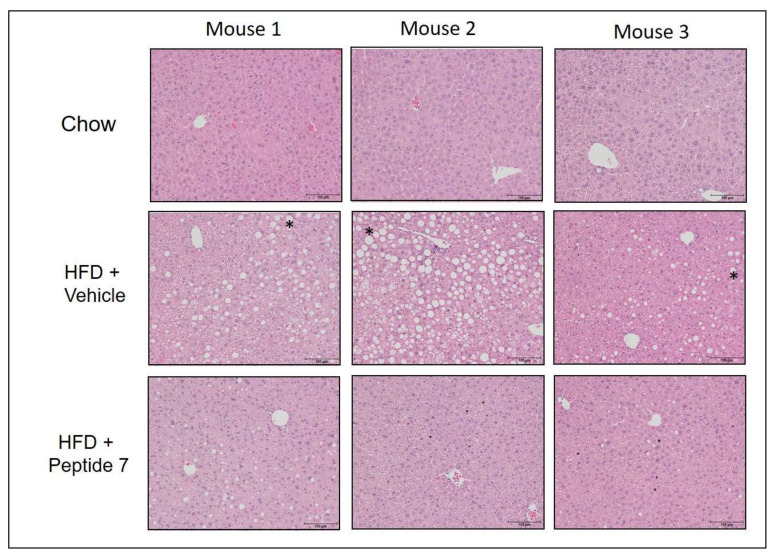
The TNFSF14 peptide, Peptide 7, reduces high fat diet-induced steatosis. Representative images are shown for *n* = 3 mice/group. * = steatotic vesicle.

**Table 1 ijms-22-10647-t001:** TNFSF14 peptides identified by computational alanine scanning as major contributors to ligand-receptor binding energy.

Peptide	Residue Numbers in TNFSF14	Sequence
1	98–117	GANASLIGIGGPLLWETRLG
2	166–180	LYKRTSRYPKELELL
3	219–228	PGNRLVRPRD

**Table 2 ijms-22-10647-t002:** Cysteine mutation analysis.

Peptide	Site 1 ^a^	Site 2 ^a^	Δ Affinity for TNFRSF14 at Site 1 ^b^	Δ Affinity for TNFRSF14 at Site 2 ^b^	Δ Affinity for LTβR at Site 1 ^b^	Δ Affinity for LTβR at Site 2 ^b^
2.1	Leu1	Leu14	−1.32	−0.70	−0.24	−0.25
2.2	Leu1	Leu15	−1.32	−1.75	−0.24	−0.82
2.3	Tyr2	Glu11	−2.37	−6.38	−1.06	−3.18
2.4	Lys3	Leu12	+6.01	+24.51	+3.77	+5.97

^a^ Peptide 2 examined; residue numbering used here commences from one, rather than following TNFSF14 residue numbering. ^b^ Refers to the Δ Affinity for mutating residue to alanine, as obtained from the computational alanine scan.

**Table 3 ijms-22-10647-t003:** Peptides selected from in silico optimisation for in vitro evaluation.

Peptide	Sequence	Δ Affinity for TNFRSF14 ^a^	Δ Affinity for LTBR ^a^	AGGSCORE
2.4 ^b^	LYCRTSRYPKECELL	-	-	58.14
5	LRCRWNRYPRECELR	+0.455	−13.966	1.492
6	LRCRWSRYPRECELR	−7.626	−7.649	2.115
7	LRCRISRYPMECRLL	−12.161	−10.728	39.79
8	LYCRTSRLPRECELR	+7.286	−10.077	44.35
9	LYCRTTRYPRICELK	−7.479	−8.686	7.35
10	LRCRISRYRYECRLL	−24.000	−20.632	24.59

^a^ Recalculated based on mutating all relevant residues of peptide 2.4 at once. ^b^ This peptide is included in this table for reference AGGSCORE.

**Table 4 ijms-22-10647-t004:** Peptides chemically synthesised for in vitro evaluation.

Peptide	Sequence	[M + 2H]^2+^ ^a^	Retention Time (mins) ^b^
5	LRCRWNRYPRECELR	1024.3 *m/z*	6.1
6	LRCRWSRYPRECELR	1011.1 *m/z*	6.0
7	LRCRISRYPMECRLL	953.5 *m/z*	7.0
8	LYCRTSRLPRECELR	946.9 *m/z*	5.9
9	LYCRTTRYPRICELK	956.4 *m/z*	6.4
10	LRCRISRYRYECRLL	999.2 *m/z*	6.9

^a^ Mass Spectra results showing the double charged adduct for each peptide. ^b^ HPLC retention time.

## Data Availability

The data presented in this study are available on request from the corresponding author.

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
