# Peer review of "TNFSF14-Derived Molecules as a Novel Treatment for Obesity and Type 2 Diabetes"

_ijms, 2021, doi:10.3390/ijms221910647_

Round 1

Reviewer 1 Report

This is an interesting study with potentially important results.  To properly interpret the in vivo findings the authors need to present additional data:  10 food intake and weight in treated treated animals.  Is this really an effect of the peptide(s) or is it a consequence of decreased food intake/weight loss owing to toxicity of the substance administered.  Ideally they should have used TNFSF14 as a control in all of these experiments.  Lastly they should demonstrate inflammatory markers in the animals treated.  Do the peptides increase inflammation/metabolic rate by fever?  CRP, sed rate and more complete cytokine panel would be needed.

Author Response

This is an interesting study with potentially important results.  To properly interpret the in vivo findings the authors need to present additional data: 

  • food intake and weight in treated treated animals.  Is this really an effect of the peptide(s) or is it a consequence of decreased food intake/weight loss owing to toxicity of the substance administered. 

We have now included food intake and weight data. Peptide 7 did not affect either parameter. Hence, it appears that peptide 7 is not toxic. Please see new red text on lines 186-188. Supplementary figure 2 is included.

  • Ideally they should have used TNFSF14 as a control in all of these experiments.

Unfortunately, we did not include full length TNFSF14 in our experiments but we did include discussion in red text on lines 358-361 highlighting that this should be done in future experiments. 

  • Lastly they should demonstrate inflammatory markers in the animals treated.  Do the peptides increase inflammation/metabolic rate by fever?  CRP, sed rate and more complete cytokine panel would be needed.

We did conduct IL-6, TNF-a (pro-inflammatory cytokines) and IL-10 (anti-inflammatory cytokine) ELISAs on serum from our mice. We did not identify any of these cytokines in serum from our mice. Hence inflammation due to Peptide 7 was no evident. We have added red text on lines 242-244 and 352-354 and lines 500-501.

Reviewer 2 Report

Review of

TNFSF14-Derived Molecules as a Novel Treatment for Obesity and Type 2  Diabetes

This is an interesting paper, the study is relevant, and the author contribute to this field of research.

The title reflects the main purpose of the manuscript precisely. The methods used and the results were well described, the conclusions are relevant.

I have only one observation, the authors need to check the list of references and correct it according to the Instructions for authors. 

Author Response

This is an interesting paper, the study is relevant, and the author contribute to this field of research.

The title reflects the main purpose of the manuscript precisely. The methods used and the results were well described, the conclusions are relevant.

  • I have only one observation, the authors need to check the list of references and correct it according to the Instructions for authors.

All references have been carefully checked and formatted according to the instructions to authors. 

Reviewer 3 Report

Authors should be congratulated for approaching an interesting topic.

Specific suggestions to be followed.

Authors should clearly state that the inner mechanisms underlying NAFLD are far from being clarified as evident in...J. Clin. Med. 2020, 9(1), 15; https://doi.org/10.3390/jcm9010015

Recent results indicate that Light (TNFSF14) deficiency in high-fat high-cholesterol diet improves hepatic glucose tolerance, and reduces hepatic inflammation and NAFL. This is accompanied by decreased systemic inflammation and adipose tissue cytokine secretion and by changes in the expression of key genes such as Klf6 and Tlr4 involved in NAFLD. These results suggest that therapies to block LIGHT-dependent signalling might be useful to restore hepatic homeostasis and to restrain NAFLD...as evident in ...Genetic inactivation of the LIGHT (TNFSF14) cytokine in mice restores glucose homeostasis and diminishes hepatic steatosis. Diabetologia. 2019 Nov;62(11):2143-2157.

Authors should present their data as means plus/minus SD and   not SEM because readers are interested in knowing the dispersion of the values and not the  precision of the mean, due to the paucity of observations for each group, i.e., four/five.

Author Response

Authors should be congratulated for approaching an interesting topic.

Specific suggestions to be followed.

  • Authors should clearly state that the inner mechanisms underlying NAFLD are far from being clarified as evident in... Clin. Med.20209(1), 15; https://doi.org/10.3390/jcm9010015

We have taken this manuscript on board for our revision. We have added red text on line 269 and added reference 19 in red text on lines 573-574.

  • Recent results indicate that Light (TNFSF14) deficiency in high-fat high-cholesterol diet improves hepatic glucose tolerance, and reduces hepatic inflammation and NAFL. This is accompanied by decreased systemic inflammation and adipose tissue cytokine secretion and by changes in the expression of key genes such as Klf6 and Tlr4 involved in NAFLD. These results suggest that therapies to block LIGHT-dependent signalling might be useful to restore hepatic homeostasis and to restrain NAFLD...as evident in ...Genetic inactivation of the LIGHT (TNFSF14) cytokine in mice restores glucose homeostasis and diminishes hepatic steatosis. Diabetologia. 2019 Nov;62(11):2143-2157.

We have taken this manuscript on board for our revision. We have added red text on lines 283-285 and added reference 21 in red text on lines 577-579.

  • Authors should present their data as means plus/minus SD and   not SEM because readers are interested in knowing the dispersion of the values and not the  precision of the mean, due to the paucity of observations for each group, i.e., four/five.

All graphical data for Figures 3, 4, 5, 6, 7, 8, 9, 10 and 11 and Supplementary figure 2 are now presented with SD. This is also indicated in red text in all corresponding figure legends and on line 503.

Round 2

Reviewer 1 Report

The authors have sufficiently addressed my concerns.  

Reviewer 3 Report

Authors improved their manuscript according to comments